# A Vertical Cooperation Model to Manage Digital Collections and Institutional Resources

**Jack Maness \*** , **Kim Pham \*, Fernando Reyes and Jeff Rynhart**

University Libraries, University of Denver, Denver, CO 80208, USA; fernando.reyes@du.edu (F.R.);
jeff.rynhart@du.edu (J.R.)

**\*** Correspondence: jack.maness@du.edu (J.M.); kim.pham60@du.edu (K.P.)

**Abstract:** The technology space of the University of Denver Libraries to manage digital collections and institutional resources isn't relegated to one department on campus – rather, it distributed across a network of collaborators with the skills and expertise to provide that support. The infrastructure, which is comprised of an archival metadata management system (Archivespace), a digital repository (Node.js + ElasticSearch), preservation storage (ArchivesDirect), and a streaming server (Kaltura) is independently but cooperatively managed across IT, library departments and vendors. The coordinated effort of digital curation activities still allows each group to focus on the service they have the most vested interest in providing. This paper will talk about the different management and development practices involved in developing our integrated infrastructure to provide digital collections as a service.

**Keywords:** digital collections; open-source software; project planning; service management; digital libraries; management models; vertical cooperation; integration architecture; third space

## 1. Introduction

The University of Denver (DU) Libraries are building an integrated, modular digital repository architecture, leveraging systems and expertise managed among many Libraries departments, campus divisions, and both proprietary and non-profit cooperative organizations. The intent is to gain more flexibility and stability in the system, while also maximizing efficiencies. While similar work employs system-integration efforts, our approach is novel in its highly collaborative management approach. The approach could prove useful for organizations of any size, but particularly for libraries and cultural heritage organizations with limited software development staff or budgets. These creative approaches to technical innovation and collaboration can be used to meet user needs.

*Institutional Background*

DU Libraries have a long history of building innovative technology in collaborative environments to meet user needs. Between 1973 and 1988, development of some of the earliest automated library systems integrated library, shared catalog, serials management, and automated interlibrary loan systems began at DU and the non-profit regional consortia at DU played a leading role in creating. These included an early integrated library system, shared catalog, serials management solution, and an automated interlibrary loan system. In 1988, a for-profit subsidiary was created to manage two of these systems, and until they were sold in 1995, one of them was used by over 400 libraries across the United States [1].

This rich collaboration and tradition of innovation continues to this day. In the 1990s and early 2000s, DU played a leading role in developing cooperative, large-scale digitization and digital object retrieval systems, most notably through the Colorado Digitization Project [2,3]. The Alliance, as the

consortia is now known, continues to support a unionized catalog for dozens of institutions in the region and has developed with members e-resource management, collection analysis, and digital repository tools and systems [4].

In 2006, member libraries in the Alliance began work on a shared digital repository leveraging open source software [5]. The project built a system hosting over 10-terabytes of data, but over the course of the following decade, when there was an economic recession and resulting decline in institutional budgets, the difficulties in meeting the needs of both academic libraries (who wished to focus on hosting research publications) and public libraries (more concerned with cultural heritage objects), and the rise of commercial turn-key solutions made it difficult to realize economies of scale, and both content and knowledge were transferred to members libraries in 2015 [6].

DU proceeded on two parallel paths with its digital content. It continued to manage the open-source stack for digitized library collections and also began a subscription to a commercial solution for faculty and student publications, in addition to supporting new services in journal publishing. Both systems proved to have benefits and drawbacks, the first being more stable and simpler to manage, the second being that it is more customizable and compliant with established archival and preservation standards and best practices.

Staffing requirements for managing the open-source system, once centralized through the Alliance, became more clearly challenging over the course of the next two years. Positions were reallocated due to the needs to manage other systems, a large reorganization of campus information technology exacerbated the challenge, and in 2018, a new approach to both systems development and structural management was developed.

Two departments in the Libraries and the portfolios for the librarians who head them, Digital Collections Services and Library Technology Services, were tasked with exploring alternatives to the existing software, and to support and provide vision for these efforts. Negotiations with stakeholders to retire dozens of existing applications, and close work with centralized campus divisions to migrate some to enterprise solutions, helped enable new development work.

Drawing on DU's history for innovation and collaborative work, the new infrastructure rebuild would emphasize modularity and flexibility. It would rely more on technical structures which are more common across other industries, rather than library- and archival- specific architectures that require developers to learn esoteric systems that may not be totally necessary. A matrix of stakeholder roles and systems guides this work (see Table 1).

To manage all of this infrastructure, our approach is modelled on vertical cooperation model approaches, a term we use very loosely, taken from value chain management theory [7]. This theory posits that business relationships belong in four categories: cooperation, competition, coopetition, and coexistence [8]. These relationships can either be vertical, meaning the relationship being described happens across firms involved in different stages of the value creation of trade, or horizontal, meaning the relationship is between similar firms, or at the same stage of value creation or trade [9].

Applied more specifically to our work—in the information, communications, and technology sector (ICT)—it is best understood as being part of the Internet of Things industry, a cross-industrial ecosystem, where it is unlikely for a single firm to provide a complete solution or product. Or, as Ghanbari et al. summarize, "companies require resources and knowledge from different fields that do not necessarily belong to a single industry. Therefore, relationships must be built within and across industries" [10].

By choosing a vertical cooperation partnership, these firms interact to combine complementary resources and knowledge towards a common goal (in the business case, the relationship is often optimized to deliver services or goods) [8]. In vertical cooperation, collaboration takes place at different stages of value creation or trade and it's seen as a more intensive collaboration that aims to optimize the relationship to deliver services or goods. In our case, with different specializations in the information technology and library sector, it is difficult to consolidate the resources and knowledge into one single department and have it be their sole responsibility. We realized that management of the

digital collections ecosystem needs to be distributed across campus, vendors, and library departments. Participation from each group encompasses the full range of activities and services required to deliver digital collections as a service.

**Table 1.** Vertical Cooperation Responsibilities.

| Role | Object Repository | Metadata Management System | Preservation | Media |
|---|---|---|---|---|
| **Owner** | Library Technology Services | Digital Collections Services | Digital Collections Services | The University of Denver (DU) IT |
| **Plan** | Library Technology Services | Digital Collections Services, Open source Community | Library Technology Services, Digital Collections Services, Open source Community | DU IT, Open source community |
| **Build** | Library Technology Services | Digital Collections Services, Open source Community | Library Technology Services, Digital Collections Services, Open source Community | DU IT, Library Technology Services, Open source community |
| **Operations (Deploy/Monitor/Upgrade)** | Library Technology Services | Alliance | Vendor | DU IT, Vendor |
| **Admin** | Digital Collections Services, Special Collections and Archives, Library Technology Services | Digital Collections Services, Special Collections and Archives, Library Technology Services | Digital Collections Services, Library Technology Services | DU IT |
| **Testing** | Library Technology Services | Digital Collections Services, State Consortium Library (Alliance) | Digital Collections Services, Library Technology Services | DU IT |
| **Repository integration** | Library Technology Services | Digital Collections Services, Library Technology Services | Library Technology Services | Library Technology Services, DU IT |
| **Documentation** | Library Technology Services | Digital Collections Services | Digital Collections Services, Library Technology Services | DU IT |
| **Stakeholders** | Digital Collections Services, Special Collections and Archives, Library Technology Services | Special Collections and Archives, Digital Collections Services | Digital Collections Services, Library Technology Services | Campus |

Vertical cooperation presents its own challenges as well. The comparison in Table 2 shows the advantages and disadvantages of the distributed vertical cooperation approach, compared to centralized management of systems. These points are drawn from prior takeaways from the collective experiences that we had in being singularly responsible for managing monolithic systems, a more traditional approach. By integrating existing systems and leveraging the expertise at the Alliance, campus information technology, and both non-profit, open-source, and commercial communities, the approach is intended to be adaptive and sustainable, and it follows a rich tradition in innovation and collaboration in technology at DU.

**Table 2.** Advantages and Disadvantages to vertical cooperation.

|  | Advantages | Disadvantages |
|---|---|---|
| **Resource management** | lower resource requirements —reduce burden of one dept having to manage entire stack | loss of control |
| **Control** | independence—economic, technological, legal. separation of concerns—department can manage the way they want to for most of their own purposes. parts can be swapped out | integration requires coordination, dependent on others to "play their part" |
| **Knowledge** | deep collective knowledge, no black boxes | knowledge is siloed per department |
| **Partner relationships** | abundant mentality towards partners—invested in each other's success | additional potential points of failure if there are departmental or political roadblocks |
| **Responsibility** | greater efficiency if responsibility is distributed | risk if responsibility for integration unclear |

## 2. Rebuild Design Process

Prior to the infrastructure rebuild, the library already had a number of systems that were used to manage the same digital objects during different stages of the curation lifecycle. Library Technology Services maintained the digital repository, which provides viewing, search, and browsing of access copies of objects, and oversees the preservation system which is for the longer term storage of original (master) files. Campus IT has a streaming media server that is used to house access copies of archival media. Digital Collections Services and our library consortium maintains the metadata management system that is used to describe both our digital and physical archival materials.

We found the phrase 'Third Space', which is used in Internet arts and culture theory by Randall Packer to describe a distributed, networked system, to be a particularly useful framing [11]. Drawing from Edward Soja's theory, which describes understanding the physical first and second mental construction of space simultaneously as the 'Third Space', Packer goes on to describe the Third Space as a "shared, social space" [12]. Packer finds that the Web is a medium primed to be a Third Space, bringing about new relationships and social dynamics.

This helped us frame our approach; the nature of our distributed web-based architecture is suitable for distributing core functions to multiple systems that excel in their particular area. As in the Third Space, it is a hybrid approach that deconstructs or brings through the cooperative nature of managing systems, transgressing boundaries to form new connections and relationships across departments. Each department manages the system where they are also a primary administrator. This means each department develops a high level of expertise over their system, and develops workflows that are optimized to meet their own curatorial needs. This, however, makes it more difficult to link together information about the same objects across systems.

For us, it is also the best way to be sustainable, given the size and resources of each individual department. Since it was desirable to maintain distributed management, the library initiated a new project to develop the integrations between systems that would automate the syncing of information throughout the ecosystem. This was preferable to taking a monolithic, single system approach, because that means that one department is never taking on the complexity of managing the one system for all digital collections workflows. We also found that no single system was able to provide the level of functionality that each separate system provided, and there was a desire and willingness to participate from all of the stakeholders, in order to preserve that functionality.

To come up with the design of our integrated ecosystem, we began with a series of user interviews and surveys to gather requirements.

From our research, we identified a set of core functions to consider what would be, for us, a successful digital collections ecosystem. At a high level, there were four requirements and we found that they mapped to existing types of library systems:

- To create and edit metadata for digital objects, organized using archival standards → Archival Resource Description System
- To search, browse, and access digital objects → Digital Repository + Media Streaming Server
- Preserve and store original (master) files → Preservation Services Framework + Storage System
- Enrich digital objects with data and contextual information from multiple systems → Microservices + Utilities

Requirements were broken down into categories, using RFC2119 [13] terminology to describe requirements as MUST, SHOULD, MUST NOT and SHOULD NOT. We emphasized developing requirements for interoperability and how the systems that we adopt must or should integrate with one another. For example, in our requirement "To create and edit metadata for digital objects, organized using archival standards", we had the following interoperability requirements:

- The system MUST support the ability to add additional archival description information to objects in a digital repository
- The system MUST provide linking information to objects available in a digital repository
- The system MUST provide linking information to objects available in a Preservation Storage System

From these requirements, we created a set of use cases for an infrastructure to manage digital collections. These use cases describe specific interactions between the user and software, a repeatable task in the system workflow. An example of interoperability use case is the ability to discover one object in a digital repository system and be able to view its additional metadata in the archival resource description system, and this can be done by a link that redirects between systems. We also decided to create granular use cases for a digital repository, knowing that we were going to develop a novel system ourselves. An example of our search use cases include performing an advanced search, browsing through paginated search results, or selecting different sort options by a particular metadata field in ascending or descending order.

Further refinements came from regular meetings each week with the development team, every other week with the project team, and every month with stakeholders to provide project updates and to work through any outstanding issues. We also met with the contractors who help with system operations to let them know what our plans for integration were, to see if there was anything else we needed to consider. We also talked to institutions developing similar infrastructure (see University of Houston, Rockefeller Archive Center, and Rutgers University). We found that these institutions had a similar modular, integrated approach that employed microservices, as well as integrating larger systems into a larger ecosystem to support digital collections.

We reached out to Rutgers University Libraries, who shared their Special Collections Reference Architecture with us. Their design is meant to increase efficiencies and economies of scale for special collections work processes across multiple library units. Their systems are separated into four functional categories: Acquisitions, Arrangement and Description, Management and Preservation, and Discovery and Reuse. These systems, using open source software (ArchivesSpace and Archivematica), vendor software (Alma and Primo) and also custom utilities, connect to each other across the categories, in order to integrate various components, to create durable links between the digital records, physical resources and their digital facsimiles [14].

Another is the Rockefeller Archive Center's Project Electron, which "aims to build sustainable, user-centered, and standards-compliant infrastructure to support the ongoing acquisition, management, and preservation of archival digital records, so that we can make them available in the broadest and most equitable way possible" [15]. One of their project values is to "support data in motion"; data constantly being passed through various systems see the need to develop an architecture for duplicate and distributed data. This led them to choose an architectural approach that allows for a whole suite of systems integrations between Archivematica, Fedora, and ArchivesSpace, "which are primarily point-to-point (a direct connection between two systems) and unidirectional (data moves in only one direction)" [16].

A third informative example is the University of Houston Libraries, who run BCDAMS (Bayou City Digital Asset Management System), an ecosystem that addresses all aspects of the digital curation lifecycle. Their goal was to implement a system that is flexible, yet comprehensive. In addition to using existing open-source systems (Hydra-in-a-Box, Archivematica, ArchivesSpace), they also developed five applications to address specific tasks relating to digital curation: Carpenters for preparing objects for digital preservation ingest, Brays for preparing objects for access ingest, Greens for the creation of persistent identifiers, Cedar for managing linked data vocabularies, and Halls for archival finding aid representation [17].

We found that these institutions share a similar modular, integrated approach that employs custom microservices and utilities, as well as integrating community-driven open-source systems into a larger ecosystem to support digital collections. We knew our strategy was going to head in a similar direction to the one these institutions took. However, we wanted to emphasize the vertical cooperation management strategy, to involve multiple departments in supporting the architecture, not just as users given the resources and size of our institution. To our knowledge, only the Rockefeller Center Archive is also taking this approach. More than the other institutions, we also wanted to focus more on the access and discovery of our digital assets. To do so, we allocated resources towards developing a custom discovery layer, that would vastly improve integration and hopefully require less modification of existing systems.

## 3. System Overview

The schematic below in Figure 1 represents the proposed digital collections' ecosystem architecture. There are four systems in the ecosystem: ArchivesSpace [18], ArchivesDirect [19], Kaltura [20], and a digital repository (digitaldu) designed in-house that addresses the needs that were not being met by the other systems.

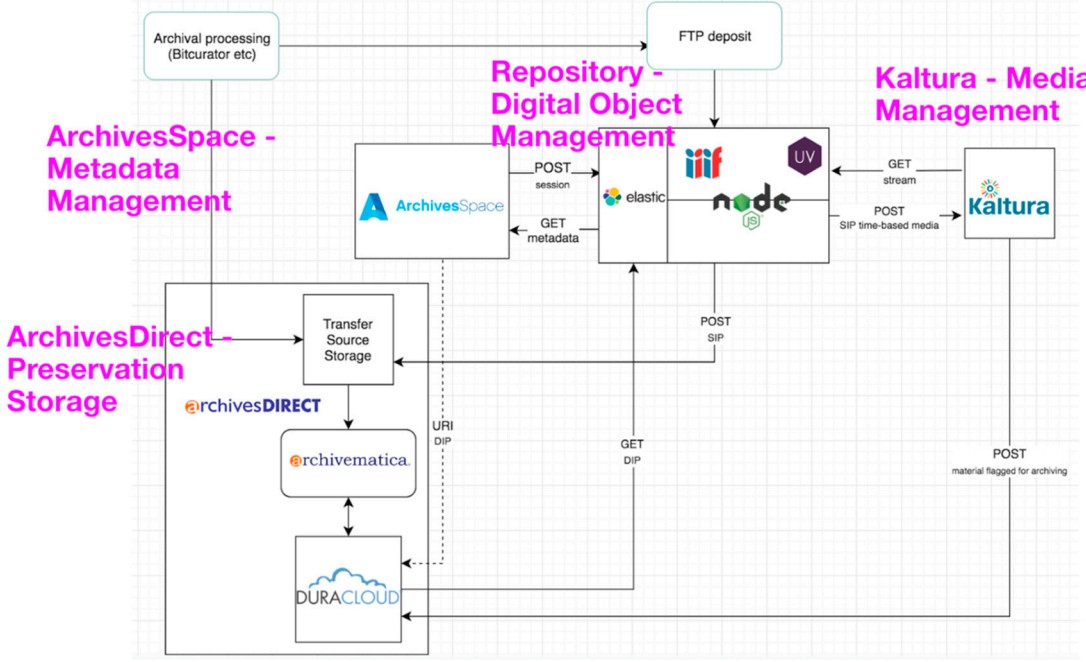

**Figure 1.** The digital collections ecosystem, an integrated infrastructure of open-source systems and microservices.

The four components operate independently, but are able to interact with one another. We developed by following the principles of web-oriented architecture, that is borrowed from

service-oriented architecture. Web-oriented architecture aims to integrate systems and users based on a core set of web protocols—in our case, everything is connected by webhooks and APIs [21].

*Digitaldu: A Digital Repository and Integration Hub*

We found that our digital repository needed to act as an integration hub for these systems to interact and to allow the user to fully explore and view our digital collections. The repository application, since it is developed locally, is the system that gave us the most flexibility to balance our digital collections workflows with the constraints of other established systems we are working with.

ArchivesSpace is one of the older systems that we maintain, and is used for archival metadata description. A plugin [22] was developed that uses ArchivesSpaces' API to retrieve metadata, then sends a JSON object to the repository to index metadata records in ElasticSearch. The ElasticSearch repository record contains information about the master file and access copies of the digital object. When an ArchivesSpace record is updated, a listener event triggers the repository indexer module to reindex into ElasticSearch.

Preservation was one of the first functions we wanted to separate out from the repository, so last year we began using ArchivesDirect, which is hosted by Artefactual and combines Archivematica and Duracloud. The formation of the archival packages and deposit involves heavy processing, which is initiated through Archivematica's API, and once it is deposited in Duracloud, it is accessed using Duracloud's API through the repository backend.

Audio and video master files are ingested through our repository and deposited into preservation storage, while access copies are ingested as a separate process into Kaltura, an audio and video streaming and delivery platform maintained by campus IT. These media are linked in ArchivesSpace, and their ID is retrieved by the repository, which then provides us with an embeddable player to stream time-based media. The player provides a number of features, such as captioning, accessibility controls, and a scrolling transcript.

The digital repository consists of a frontend (digitaldu-frontend) and backend (digitaldu-backend) application. The backend [23] is built with Node.js along with Express.js, to handle and route HTTP requests. Node.js was chosen for a few reasons—Library Technology Services started using it two years ago before and found it to perform well in our Linux environment. It is well suited to build out our event-driven microservices, which lets the repository receive and transmit messages between systems. In our implementation, we use Redis to manage and synchronize object states throughout the digital object lifecycle. This means that components can be loosely coupled and work for an asynchronous environment, such as our own. We also found that Node.js is easier to maintain compared to previous web frameworks we used in the past. Node.js automates how to resolve any version or code conflicts between dependencies, so it does not break on us on every update.

The backend is composed of eight modules:

1. Auth—admin access control, we use an LDAP authentication service
2. Dashboard—backend portal to display stats, collections, search, users, imports, import status (see Appendix A, Figure A6 for the screenshot)
3. Import—import functionality—initiates/stops/queues/logs transfers
4. Indexer—writes to SQL database, which writes to ElasticSearch
5. Repository—functionality to edit collections and digital objects (see Appendix A, Figure A7 for the screenshot)Search—backend search controls (both public and private records)
6. Stats—repository statistics, database query logic
7. Users—ability to add and remove users (see Appendix A, Figure A8 for the screenshot)

The frontend [24] is also built in Node.js with Express.js. All of the dynamic content is powered by ElasticSearch and uses custom templates for each object type. Digital objects are also IIIF resources which are rendered in UniversalViewer, except for audio and video files, which can also be rendered in Kaltura. The frontend is composed of four main modules:

1. Discovery—frontend interface controls for collections, objects, pagination (see Appendix A, Figures A1 and A2 for screenshots)
2. Search—search functionality, facets, filtering, query construction (see Appendix A, Figure A3 for the screenshot)
3. Views—digital collections page templates (see Appendix A, Figures A4 and A5 for screenshots)
4. API—creation of custom endpoint to programmatically access digital collections as data

## 4. Considerations and Future Directions

About 1.5 years into this integration project, we have learned some lessons and encountered a number of interesting technical challenges. Our approach (as expected) did not reduce the required resources for maintenance; rather the responsibility is distributed across multiple partners. It is difficult to test the ecosystem until multiple integrations are finished. There is therefore a lot of coordination and upfront setup to route events, in order to see if data is flowing through each system correctly. Systems do not perform at the same speed, so we were surprised by how much time we needed to spend to account for timeouts, throttling and queuing transfers, and figuring out ways to restart processes if they have been stopped throughout the workflow. Red Hat published an example [25] depicting some of these potential concerns of distributed architecture.

We found that monitoring is crucial in an integrated system architecture. In the backend, we developed a dashboard that regularly checks connectivity and that systems are operational. It also tells us whether or not the digital object upon ingest has the associated data that you expect it to contain. One expected constraint is having to work within each system's limitations and what each system can do with the API they provide. An additional risk is if a system's data model changes in a new version or complete rewrite. We have yet to develop a way to monitor these changes, but rather test this with each update or upgrade of a system.

There are also several managerial considerations to the cooperative model, most of which were anticipated and are articulated in Table 2. There are many staff members involved in publishing digital collections in the system, and they need adequate knowledge of, and documentation for, working in at least two of the modules, sometimes three. The portfolio for various leadership positions is also a consideration: part-way through the development project, the Digital Collections Librarian accepted a position at another institution. This departure required us to engage in a process to determine the skills and experience necessary for the position as usual, but with an added consideration with respect to the department's responsibility outlined in the cooperative model. This challenge presents itself across the cooperation, of course—the reality is that each individual in the project is an important part of its success—so maintaining skills and making implicit knowledge explicit through careful documentation that can facilitate training is crucial. It is also important to reiterate the rationale of the project with stakeholders frequently.

Both technical and managerial considerations in the vertical cooperation approach fundamentally involve challenges that are inherent to their relative opportunities: where distributing responsibility presents an opportunity for the more efficient use of skills, for example, it also presents the challenge of losing those skills and having less control over replacing them; similarly, where cooperation allows for relationship-building that can be useful beyond the immediate needs of the project, it also introduces political and social challenges that may not have otherwise existed.

When considering adopting the approach, then, an organization must carefully evaluate the relative advantages and disadvantages to it. If resources allow for a more centralized management of a monolithic system, perhaps it is more strategic to do so. However, if resources need to be streamlined, a vertical cooperation may be preferred. Irrespective of resources, however, DU's experience, following our initial release of the digitaldu backend and frontend in the Summer of 2019, and as we prepare for a second release in the Spring of 2020, the express goal of the release being improving integrations between these systems, is that a cooperative, integrated effort allows for greater innovation. Archivists,

technologists, usability experts, and developers have worked together to build a system that meets our organizational strategies and user needs, in ways we may not have otherwise been able to meet.

## 5. Future Directions

Our initial release of the digitaldu backend and frontend came out in the Summer of 2019. We began migration to this system shortly thereafter. A second release is planned for Spring 2020, with the express goal of improving integrations between these systems. In the future, we hope to have more conversations with the wider Open Repositories community around integrated, flexible, modular architectures, and in doing so, start to map out common patterns in the way these systems are being designed. If we can talk about common protocols or standards that could be used for integrations, this could influence community solutions to focus more on developing systems that can be easily integrated with one another. Institutions would have an easier time adopting these systems and feel confident in committing resources, if they could follow an architectural pattern that has been vetted and proven to work in the past.

**Author Contributions:** All authors contributed to all sections of this article. Conceptualization, background research and manuscript, J.M. & K.P. Project management K.P. Implementation, K.P., F.R. & J.R. All authors have read and agreed to the published version of the manuscript.

**Funding:** This research received no external funding.

**Acknowledgments:** The authors would like to thank Kevin Clair at Pennsylvania State University Libraries for his significant contribution in the conceptualization and implementation of this work. The authors would also like to thank Patrick Galligan, Hillel Arnold, Hannah Sistrunk, and Bonnie Gordon at the Rockefeller Archive Center, Chad Mills at Rutgers University Libraries, Andrew J. Weidner, Anne Washington, Scott Watkins, and Bethany Scott at the University of Houston Libraries, Kelli Babcock at the University of Toronto and Andrew Woods at LYRASIS for their consultation and advice.

**Conflicts of Interest:** The authors declare no conflict of interest.

## Appendix A

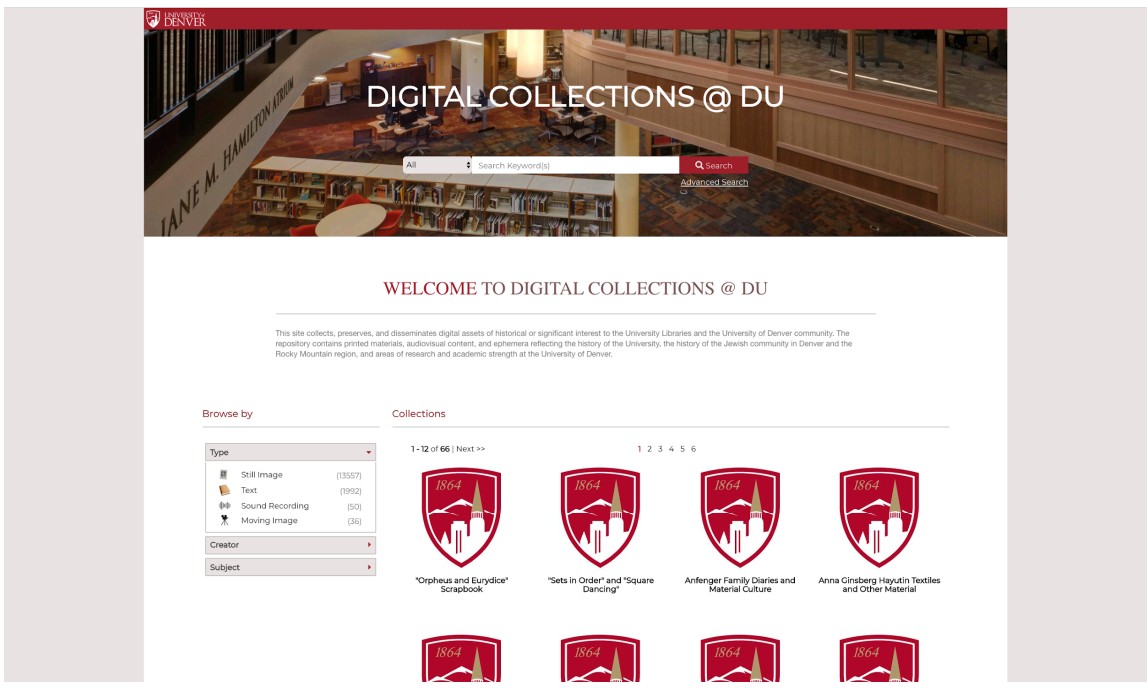

**Figure A1.** digitaldu—frontend repository screenshots (Landing page).

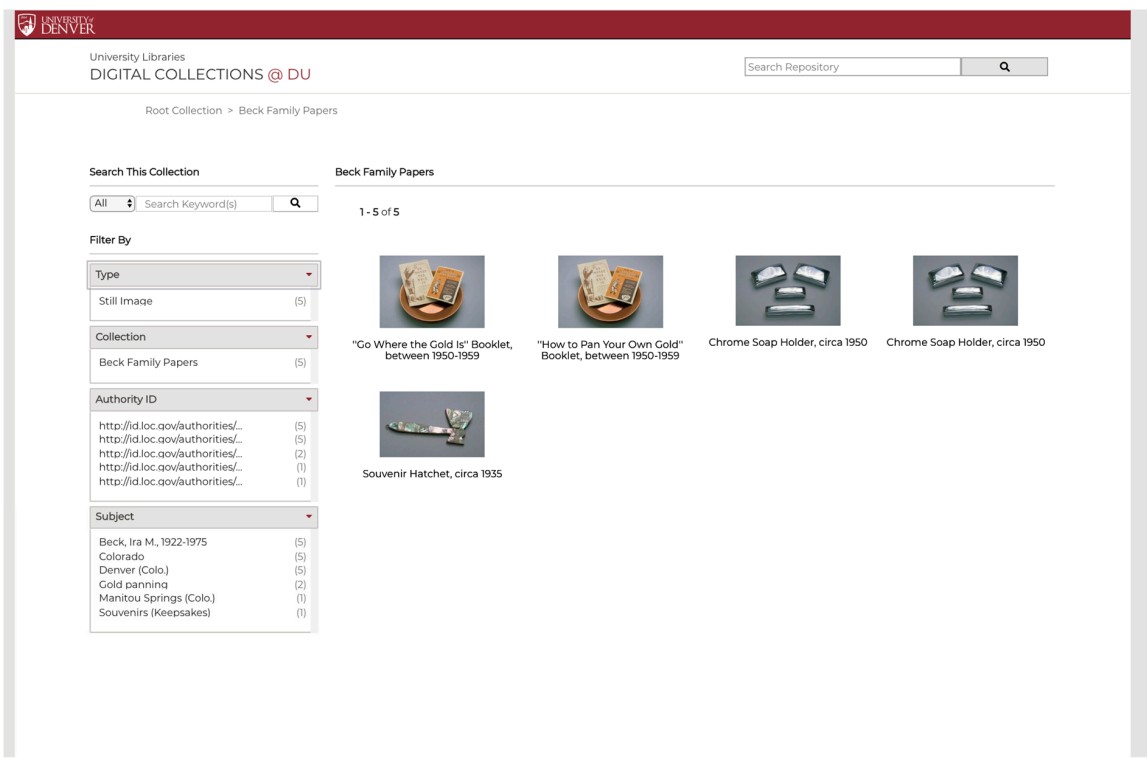

**Figure A2.** Collection view.

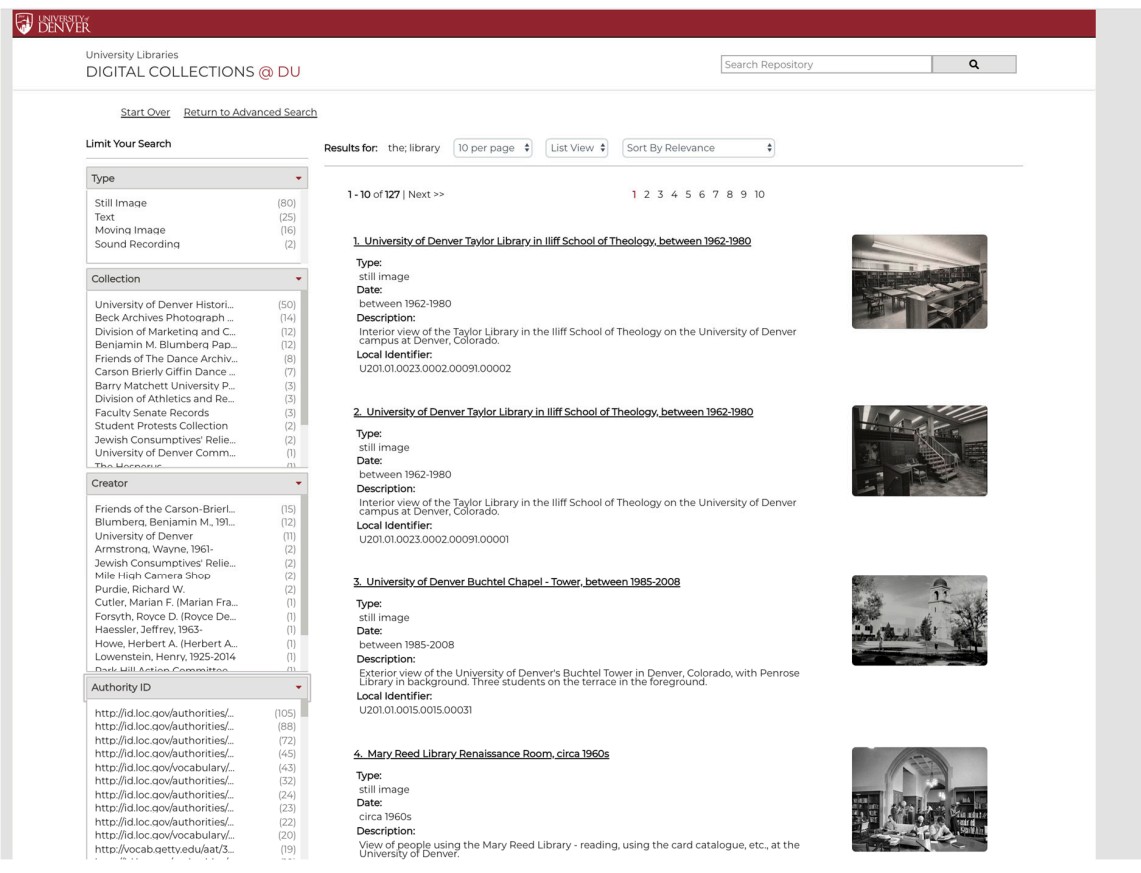

**Figure A3.** Search.

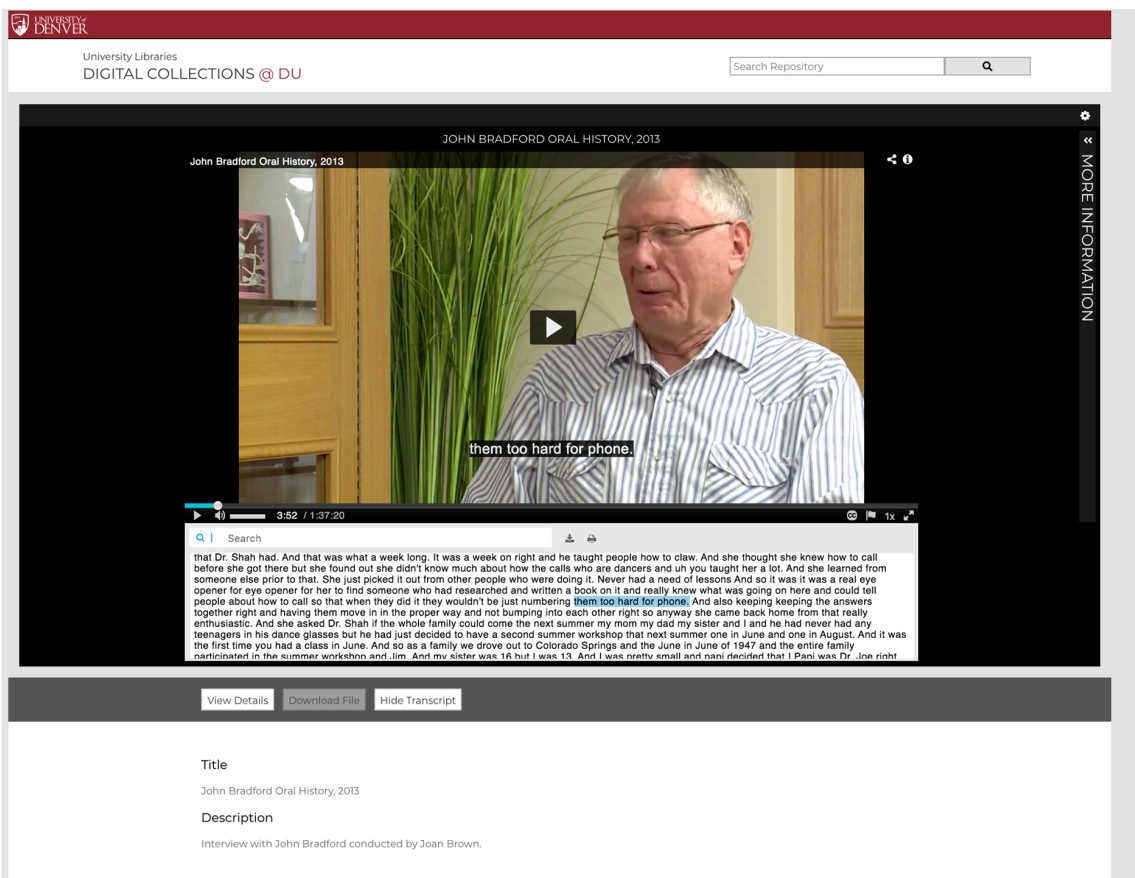

**Figure A4.** Video viewer (Kaltura embedded in UniversalViewer).

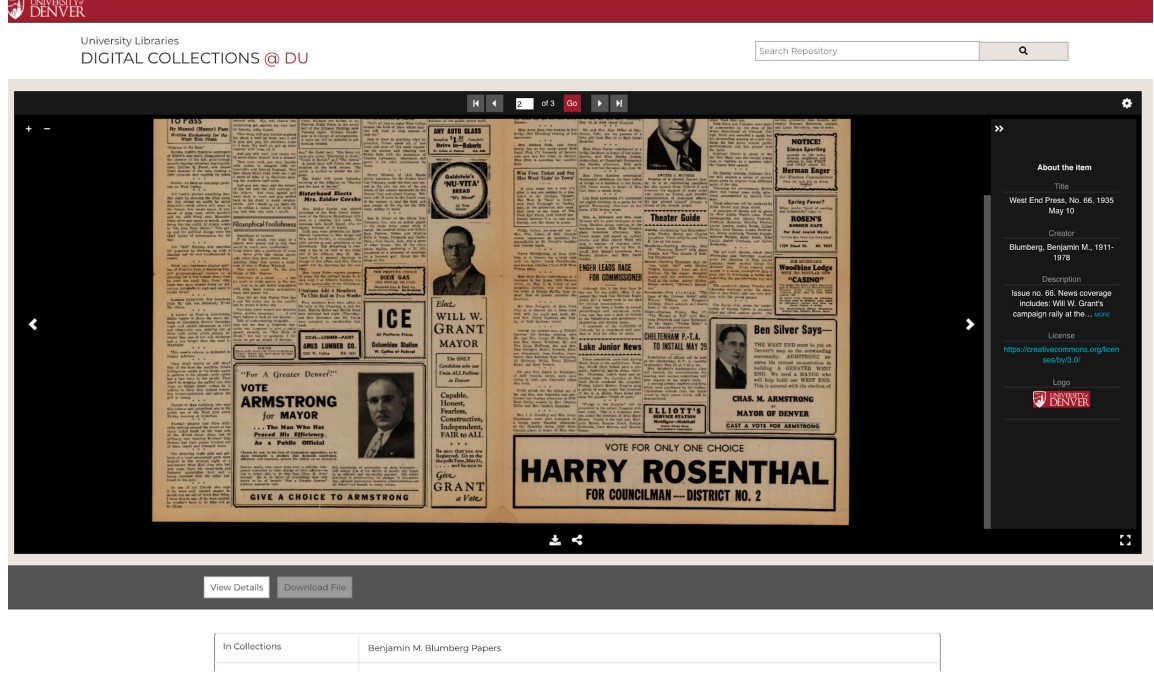

**Figure A5.** Image viewer (UniversalViewer).

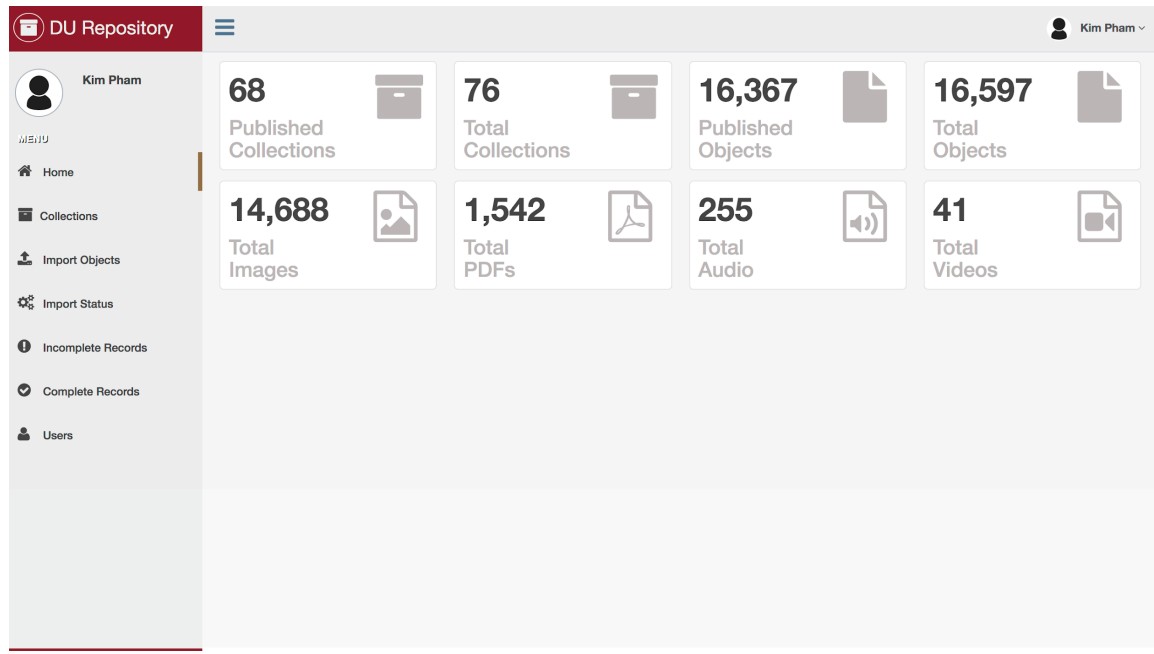

**Figure A6.** digitaldu—backend screenshots (Dashboard module).

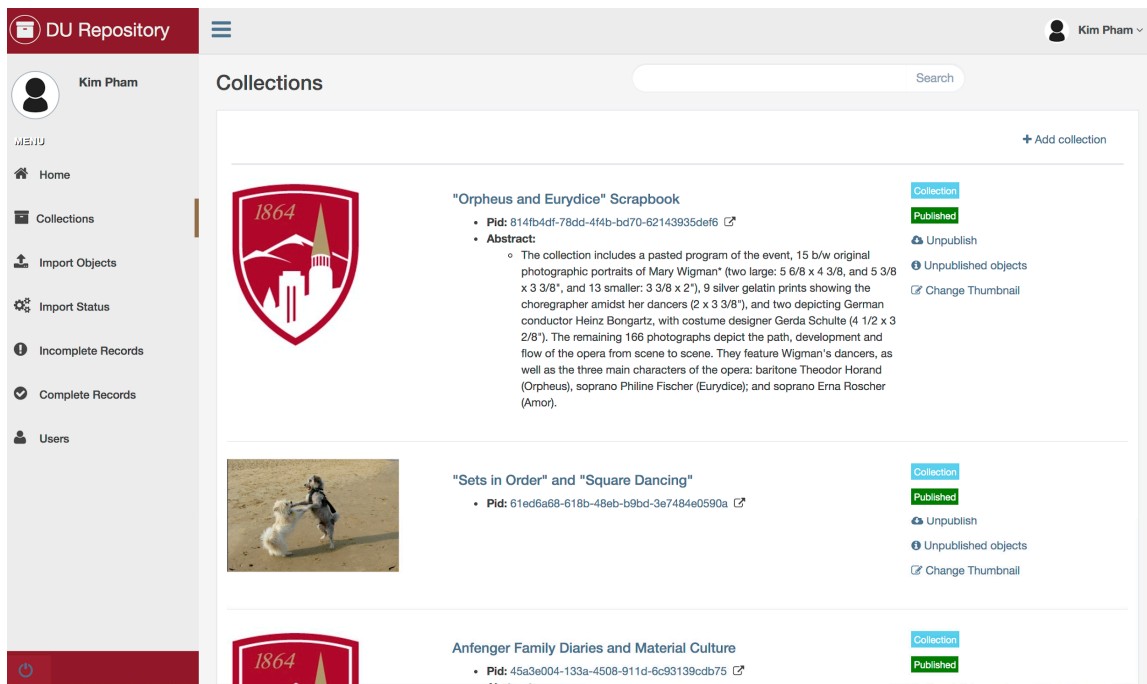

**Figure A7.** Repository module.

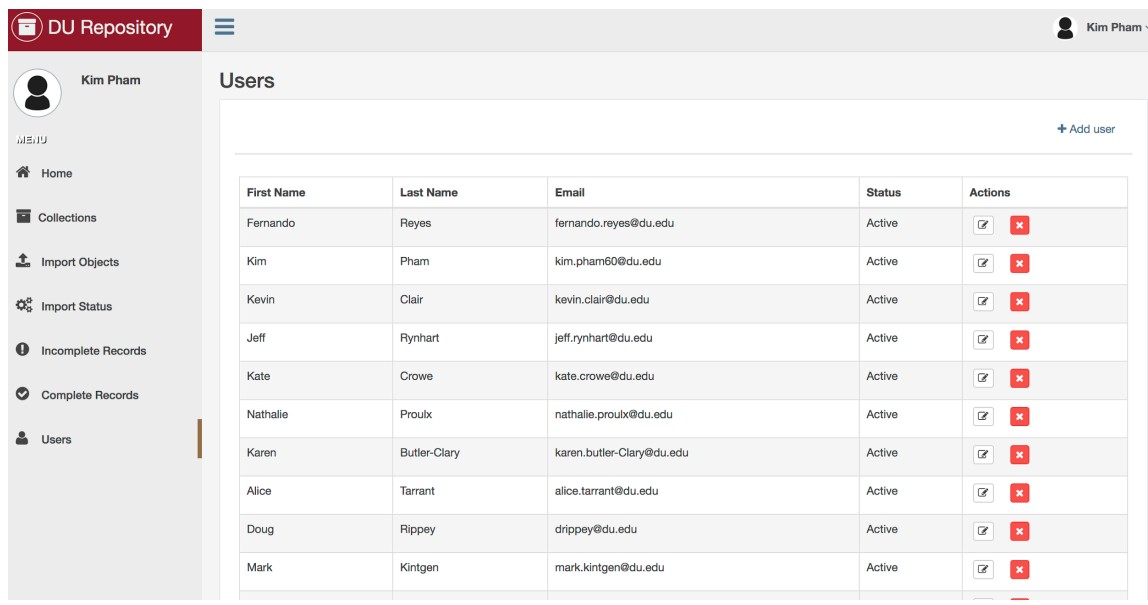

**Figure A8.** Users module.

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
