# Peer review of "A Vertical Cooperation Model to Manage Digital Collections and Institutional Resources"

_publications, doi:10.3390/publications8020023_

Round 1

Reviewer 1 Report

This is a detailed and informative report on technical developments at DU. Presentation is clear, and I believe the information could be useful to readers involved in the development of library technical infrastructure.

I have three main suggestions for revisions, as well as some minor line-by-line suggestions.

  1. I think it is essential that a clear introduction is added (either as a paragraph, or ideally a short Introduction section). As it stands the reader is thrown straight into a detailed history of technical developments at DU, without any clear sense of where the article is going, or why this material is relevant.
  2. The article could more clearly articulate the ways in which the DU approach is novel. While the steps taken and the rationale for decisions are described clearly, it would be helpful for the reader to be shown the ways in which the approach is innovative. You mention other institutions developing similar infrastructure - I think this could be expanded upon to include comparative reflections on the DU context.
  3. Related to 2., I think the applicability of the DU solution to other institutions could be discussed in more detail. In other words, I think you need to clearly state throughout how the content of this article could be used by librarians at other organisations. 

In addition, I have the following line-by-line comments"

L26 - should be either "member libraries" or "members' libraries"

L30-32 - "one ... the other ... " Consider saying "the first ... the second ... " to avoid confusion about which systems you are referring to.

Table 2 - No sources are given for the content of this table. Are the advantages / disadvantages based on your own experience/analysis, or the existing literature? If the former, then I would expect to see some description of how these were identified. If the latter then obviously references are required. 

76 - I'll leave this up to you, but I'd be interested in hearing more about the 'third space'. Could Packer's conceptualisation be introduced in more detail? This would help the reader understand how and why it influenced your thinking.

87-88 - You mention conducting exploratory research to establish user needs, but this is rather glossed over. It would be interesting to hear more about this. Perhaps you could present some example use cases?

190-191 - The last line of the article feels rather throw-away. If a discussion of common protocols and standards are important then you should say why, and more confidently assert the value of further discussion. 

Reviewer 2 Report

I think, this is an excellent report.  It provides the background, evidence, support for conclusions, and a sound methodology.  Anyone familiar with the Colorado Digitization Project (CDP) will recognize the vision behind Digital DU that made CDP such a compelling model.

Digital DU is also a model for academic institutions  to invest in their digital infrastructures and heritage collections.

You will need to go through spell-checking: line 31: it should be compliant, not complaint.  Lines 57-61.  Sentences starting with and and but usually end with an exclamation mark, so you want to replace those with However; and Also; or restructure your paragraphs.

I feel that the intro could give away some of the main concerns and intentions in this report.  The background becomes the introduction, which is surely building up to where you introduce vertical cooperation as the key concept for your report.  This is the concept that makes your institution function so effectively. Please show what other institutions that are not getting it (or are not interested) should be able to do. 

Reviewer 3 Report

The article reports on a University of Denver Libraries initiative to restructure digital collections infrastructure in a way that distributes responsibility for different aspects of the system to different units, through a vertical cooperation model. 

The description of the institutional background, the work that was undertaken, and the resulting integrated system is clear and succinct, and will likely be useful to other libraries as they seek to restructure. The authors clearly explain the decisions made, and articulate the technical challenges and workflows well.

I recommend the following changes to make this article more useful to a broader base of readers:

  1. In the introduction, explain what the intended contribution of this paper is: How does this publication aim to help other libraries or information organizations? How would DU's experience generalize to other institutions, or what are the main takeaways? 
  2. Push for greater clarity of key concepts.
    1. The definition of "vertical cooperation" is unclear. It would be most helpful if the authors could articulate what the alternatives to this model are, give clearer examples of how this model works, and how the alternatives differ. It would be helpful if the authors could provide more information about how other institutions have implemented this model or not, particularly since the paper suggests that the model is innovative. How is it innovative? And how does it differ from previous implementations the authors cite at Houston, Rockefeller, and Rutgers?
    2. Table 2 implies that the vertical cooperation model has advantages and disadvantages, but relative to what? 
    3. The "third space" theory (in Rebuild Design Process) sounds very interesting but is given minimal explication. Please provide more context and relate it more clearly to the design process.
  3. The "Considerations" section is probably the most generally useful section for other institutions. It should provide a clearer set of takeaways or implications for other institutions to learn from this experience, including summarizing the main takeaways or lessons learned that pop up through the rest of the article.
  4. Table 1 is a little hard to read; consider adding punctuation between each division name within a cell.
  5. There are minor grammatical and spelling problems throughout. Do a close read for these details. E.g., "which lets the repository receives and transmits messages" <-- typos/plurals. And "maintenance hasn't been too bad" <-- too informal and vague. And "a new approach both systems development and structural management" <-- missing word. Etc.
